# Estimating Wildlife Tag Location Errors from a VHF Receiver Mounted on a Drone

**André Desrochers** [1],*, **Junior A. Tremblay** [2], **Yves Aubry** [3], **Dominique Chabot** [4], **Paul Pace** [5] and **David M. Bird** [6]

1   Département des Sciences du Bois et de la Forêt, Université Laval, 2405 rue de la Terrasse, Québec, QC G1V 0A6, Canada
2   Environment and Climate Change Canada, Science and Technology, 801-1550 avenue d'Estimauville, Québec, QC G1J 0C3, Canada; junior.tremblay@canada.ca
3   Environment and Climate Change Canada, Canadian Wildlife Service, 801-1550 avenue d'Estimauville, Québec, QC G1J 0C3, Canada; yves.aubry@canada.ca
4   droneMetrics, 7 Tauvette Street, Ottawa, ON K1B 3A1, Canada; dominique.chabot@mail.mcgill.ca
5   Defense Research and Development Canada, 3701 Carling Avenue, Ottawa, ON K1A 0Z4, Canada; paul.pace@rogers.com
6   Avian Science and Conservation Centre of McGill University, c/o 10980 Dunne Road, North Saanich, BC V8L 5J1, Canada; david.bird@mcgill.ca
*   Correspondence: andre.desrochers@sbf.ulaval.ca; Tel.: +1-418-656-2131

**Abstract:** Recent studies have demonstrated the high potential of drones as tools to facilitate wildlife radio-tracking in rugged, difficult-to-access terrain. Without estimates of accuracy, however, data obtained from receivers attached to drones will be of limited use. We estimated transmitter location errors from a drone-borne VHF (very high frequency) receiver in a hilly and dense boreal forest in southern Québec, Canada. Transmitters and the drone-borne receiver were part of the Motus radio-tracking system, a collaborative network designed to study animal movements at local to continental scales. We placed five transmitters at fixed locations, 1–2 m above ground, and flew a quadrotor drone over them along linear segments, at distances to transmitters ranging from 20 m to 534 m. Signal strength was highest with transmitters with antennae pointing upwards, and lowest with transmitters with horizontal antennae. Based on drone positions with maximum signal strength, mean location error was 134 m (range 44–278 m, n = 17). Estimating peak signal strength against drone GPS coordinates with quadratic, least-squares regressions led to lower location error (mean = 94 m, range 15–275 m, n = 10) but with frequent loss of data due to statistical estimation problems. We conclude that accuracy in this system was insufficient for high-precision purposes such as finding nests. However, in the absence of a dense array of fixed receivers, the use of drone-borne Motus receivers may be a cost-effective way to augment the quantity and quality of data, relative to deploying personnel in difficult-to-access terrain.

**Keywords:** radio-tracking; Motus; drone; boreal forest; precision; accuracy; response surface; forêt Montmorency

## 1. Introduction

For several decades, radio-tracking has proven itself as a valuable tool for the investigation of animal movements at a wide range of temporal and spatial scales [1]. The advent of the Global Positioning System (GPS) and similar satellite-based systems has revolutionized wildlife tracking by transmitting precise coordinates directly from tagged animals. However, GPS transmitters remain costly and too heavy for small animals. As a result, conventional, non-GPS, transmitters remain the

predominantly used tracking technology in studies of small songbirds and many other terrestrial applications. With the exception of light-sensitive geolocators, which require recaptures to retrieve data stored in memory [2], the location of non-GPS transmitters usually has to be inferred from signal detection, signal strength, compass direction or combinations of those.

To get precise estimates of the location of animals fitted with non-GPS transmitters, field biologists will often resort to triangulation or homing, i.e., getting progressively closer to the focal animal [3]. In rugged, difficult-to-access terrain, the latter two approaches are labor-intensive and may be well outside research budgets, not to mention ever-increasing concerns about safety [4]. For species ranging over kilometers or more, the use of conventional aircraft such as Cessna planes is often the only way to obtain sufficient amounts of data, but such campaigns also pose a safety risk, with aviation accidents determined to be the leading cause of mortality among wildlife workers in the United States from 1937–2000 [5].

The advent of drones in the civil sector may offer immense potential for combining affordability, safety, and accuracy in the effort to document movements of animals fitted with non-GPS transmitters. Chabot and Bird [6] provide a review of recent advances in the use of drones for wildlife applications. They point out that despite its potential, drone-borne wildlife radio-tracking remains underdeveloped, possibly due to enduring skepticism about its potential and/or technological and operational barriers. To this day, the published literature suggests that, with few exceptions, the subject continues to be largely approached as an engineering curiosity more so than an endeavor by those who stand to benefit from its development: wildlife researchers and managers [7–10].

One of the concerns that needs to be addressed to promote the effective use of drones in wildlife radio-tracking is accuracy. Signal power density is proportional to the inverse square of the distance between the transmitter and the receiver. In principle thus, knowing the strength and direction of a transmitter signal from two locations, sufficiently distinct in space and sufficiently close in time, should yield highly accurate positions. However, transmitter signals are dampened by trees, rocks, etc., to varying extents, and may exhibit multi-path effects, making it practically impossible to infer transmitter locations from two locations with intervening obstacles.

We estimated transmitter location errors with fixed-location "test" transmitters and a drone-borne receiver in a hilly and dense forest composed mostly of balsam fir (*Abies balsamea*). We used a simple quadratic, two-dimensional response surface of signal strength against Latitude and Longitude.

## 2. Materials and Methods

We conducted this study in September 2016 at Forêt Montmorency (47.4 N, 71.1 W), a teaching and research forest located north of Québec City, Québec, Canada. The study area is a dense balsam fir/white birch (*Betula papyrifera*) boreal forest landscape with altitudes ranging from 750 to 1000 m, covered by a dense network of forestry roads (for details see [11]). We conducted seven flights, within two sectors each covering ~0.2 km$^2$, each composed of a matrix of old, mid-successional, and early-successional balsam fir stands resulting from clearcutting (Figure 1). Tree height in the study site varied from ~4 m to 12 m, with ~2500–10,000 stems/ha.

Each flight was performed by a custom-built heavy-lift quadrotor drone based on a Gryphon Dynamics airframe (Daegu, South Korea) and a Pixhawk flight controller (3D Robotics, Berkeley, CA, USA), with a payload capacity of about 4 kg including the battery. The drone was programmed to fly at a fixed altitude of 50 m above the ground at the location where it was launched and a forward speed of 5 m/s. We automated flights from takeoff to landing, and monitored them from the ground using a tablet computer. We mounted a radio receiver system on the ground-facing side of the drone. The radio receiver system was composed of a hanging omnidirectional dipole antenna attached to a weight at the bottom end, and coupled to a Funcube Pro+ dongle. The dongle was connected to a BeagleBone computer programmed to monitor and record signals simultaneously from multiple transmitters (for details see [12]).

Before each flight, we deployed five radio-transmitters at distances ranging from 18 m to 507 m from one another (Figure 1). We placed each transmitter in a tree at ca. 1.3 m above ground, with the antenna pointing up, down or horizontal. Transmitters were avian nanotags model NTQB-4-2, Lotek Wireless Inc., Newmarket, ON, Canada). Each nanotag had a unique set of pulses delivered each 5 s at a frequency of 166.38 MHz (VHF; very high frequency), a standard used by the Motus Wildlife Tracking System [12,13].

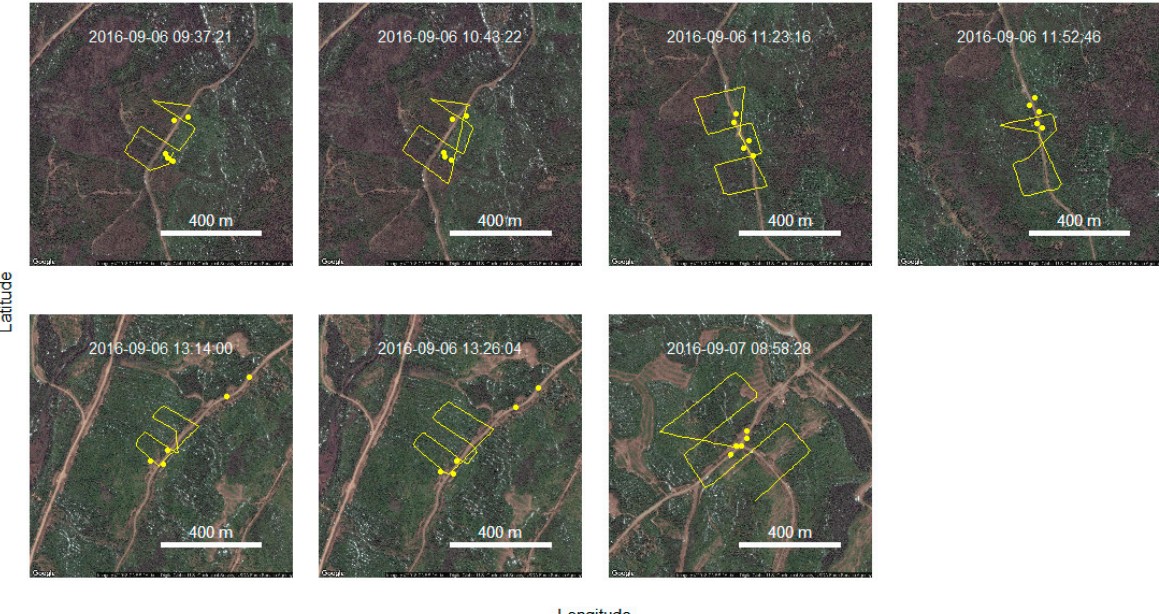

**Figure 1.** Flight transects and test transmitter locations, in yellow. Takeoff times are indicated.

After the completion of each flight, we downloaded data from two sources: the telemetry receiver and the drone navigation log. Timestamps from the two sources were synchronized, allowing us to match each signal reception from test transmitters to the exact location of the drone and in turn, the distance between the transmitters and the receiver.

As pointed out earlier, several factors may influence signal reception and strength. However, on average, signal strength should provide an unbiased estimate of the distance between the transmitter and the receiver. Thus, a two-dimensional array, i.e., a map of signal strengths, should inform us about the true location of the transmitter. We estimated the location errors with two methods. First, we retained the drone location at which the signal was strongest, and calculated the Euclidean distance between the drone XY coordinates, and the XY coordinates of the source. Second, we modelled signal strength as a function of the drone's XY coordinates in meters from a Modified Transverse Mercator map projection (Easting and Northing) using two quadratic functions:

$$\text{Easting: Dbm} \sim \beta_0 + \beta_1(\text{Easting}) + \beta_2(\text{Easting}^2) + \varepsilon \tag{1}$$

$$\text{Northing: Dbm} \sim \beta_0 + \beta_1(\text{Northing}) + \beta_2(\text{Northing}^2) + \varepsilon \tag{2}$$

where Dbm is the signal's strength, $\beta_i$ regression estimates, and $\varepsilon$ a vector of model residuals. The formulas yielded a peak signal strength when the regression estimate for the quadratic term was negative. Note that in the presence of a peak signal strength both on X and Y coordinates, only *relative* signal strength will be required to estimate transmitter locations. Differences in signal strength among

transmitters, whether because of manufacturing or placement in the forest, are measured by the models' intercept ($\beta_0$). We obtained Easting and Northing estimates by:

$$\hat{E}, \hat{N} = \frac{-\beta_1}{2 \cdot \beta_2} \tag{3}$$

We conducted all analyses with the statistical software R version 3.5.0 [14].

## 3. Results

We obtained 669 detections of the test transmitters from the combined drone flights. Signal strength decreased significantly with increasing distance to drone, with the furthest detection at 534 m (Figure 2, $F_{1,646} = 181.7$, $p < 0.001$). The orientation of the transmitter's antenna also had a significant effect on signal strength ($F_{2,646} = 62.6$, $p < 0.001$), with antennae pointing upward performing best.

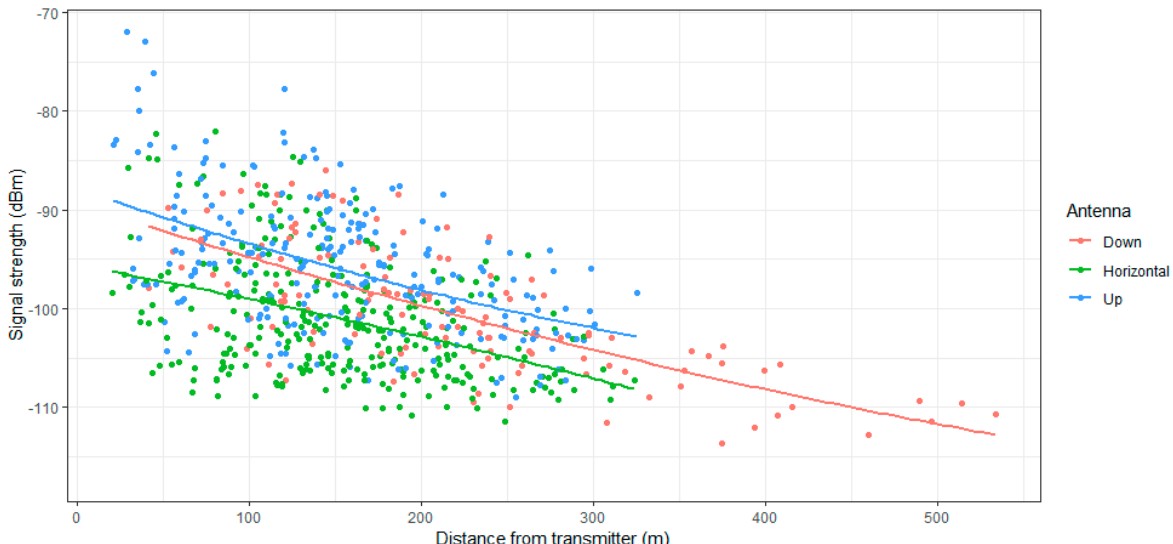

**Figure 2.** Signal strength in response to the distance between the drone and the transmitter.

Given that we conducted seven flights, each with five test transmitters deployed, we obtained 35 detection sets. Five to 42 detections were obtained for each detection set (mean = 19.9). Based solely on drone locations when signal strength was greatest, mean location error was 132.3 m (range: 28.8–294.2 m, n = 35). The reliability of the maximum strength method is questionable when the strongest signal is on the periphery of the drone route, because in those cases the transmitter was likely outside of the range covered by the drone. To prevent this, we removed all detection sets where the strongest signal came from a drone location on the periphery of the convex hull enclosing the detection set. The resulting subset of data yielded a mean location error of 134.0 m (range: 43.9–278.0 m, n = 17).

Nearest distances to transmitters yielded strongest signals in only two of the 35 detection sets. Furthermore, signal strength did not always increase nearer transmitters (Figure 3), leading to only seven cases where quadratic regression coefficients of signal strength against X or Y coordinates were negative, i.e., leading to a maximum estimated signal strength as required for position estimation.

Of the seven cases with estimable positions, we dropped one case with an estimate error (3034 m) greater than the maximum known distance between the drone and the transmitter (534 m). Figure 4 illustrates the remaining six cases where quadratic curve-fitting yielded estimable positions.

The mean location error from the quadratic method was 69.9 m (range 20.8–161.3 m).

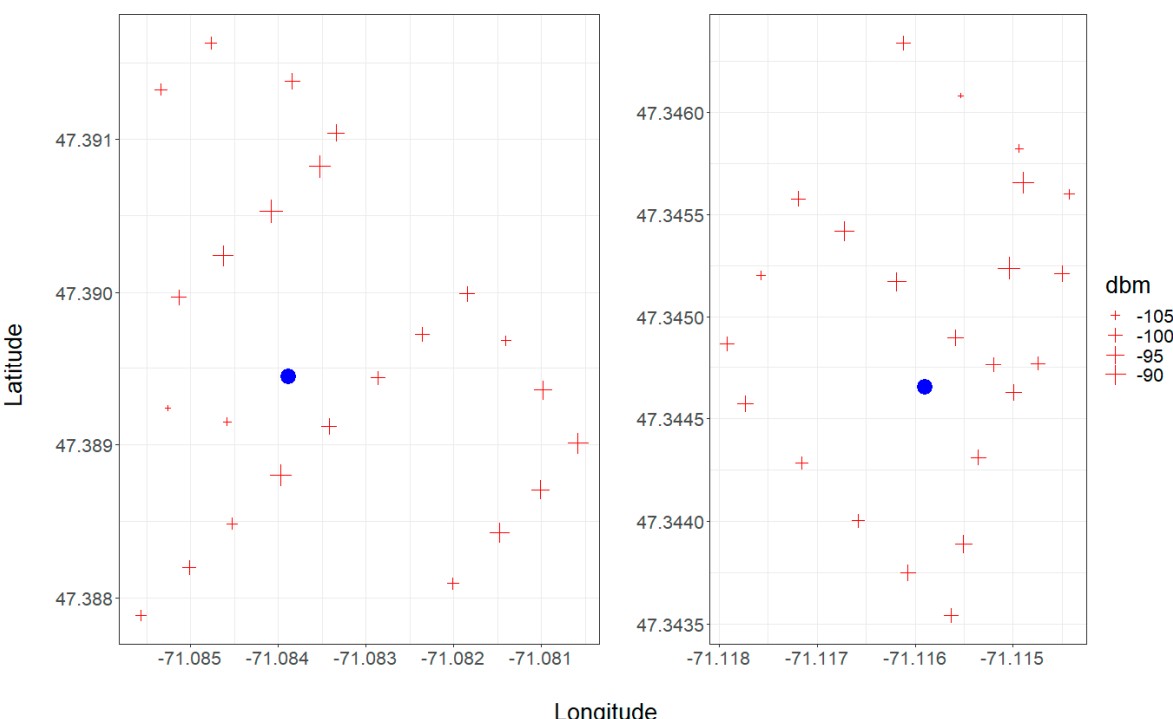

**Figure 3.** Left: best case scenario, with signal strength, depicted by cross size, roughly increasing toward the location of the test transmitter (blue dot). Right, worst case scenario, with signal strength showing no obvious relationship with distance to transmitter.

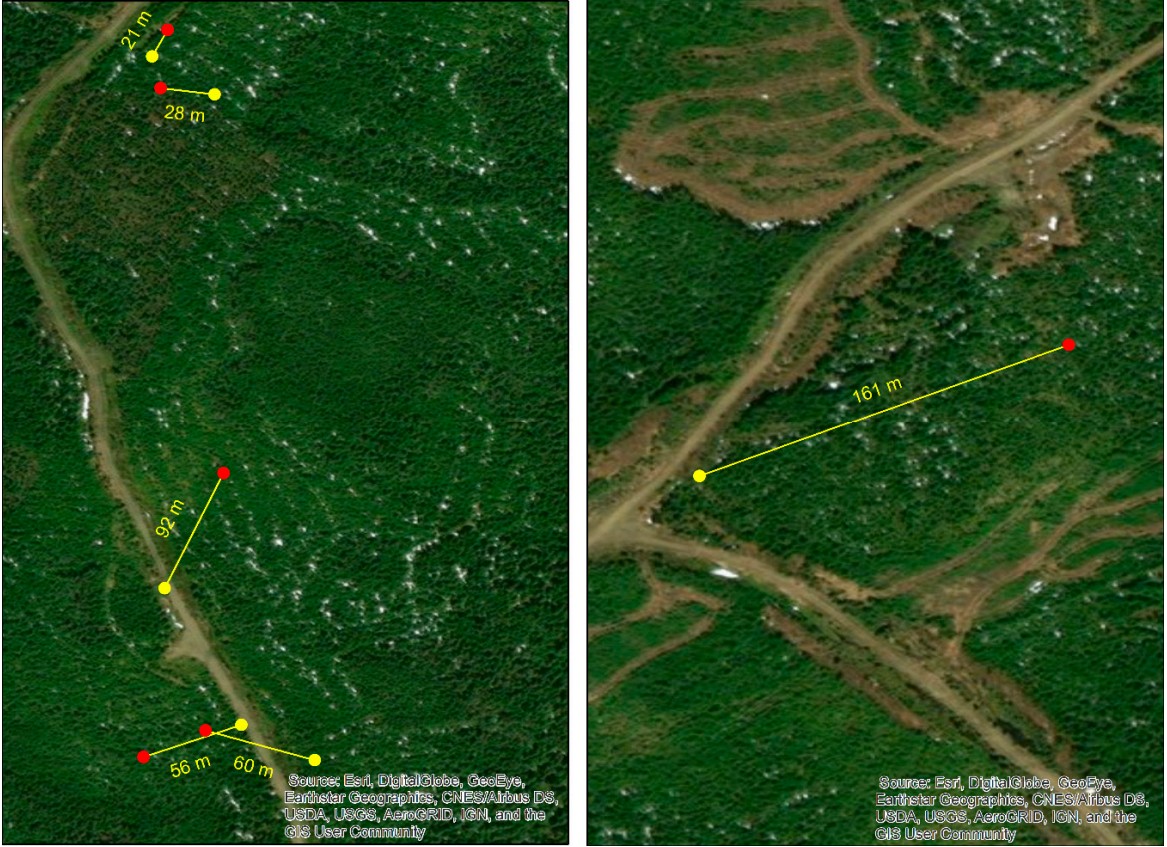

**Figure 4.** Transmitter location errors from quadratic regression estimation, based on signal strength. Test transmitter locations in yellow, estimated locations in red.

## 4. Discussion

The main outcome of this study was that drones offer an alternative to the more labor-intensive, traditional approaches for radio-tracking small birds, amphibians, or small mammals in rugged terrain. All five transmitters were detected on each of the seven flights, thus for the purposes of simple detection at a range of a few hundred meters, a drone appears highly effective. However, the precision of the detection-by-drone method is likely insufficient for finer-scale applications such as finding nests or dens or documenting microhabitat use. We explored two ways to estimate source locations, based on observed vs. modeled maximum signal strength. Observed maximum signal strength has the benefit of being simple and easily obtainable (larger sample), but it wastes the bulk of the information obtained by the drone. Furthermore, it is overly sensitive to outliers. The estimation method based on quadratic estimation has the advantage of combining a comprehensive use of the data with computational simplicity. In the present study, the quadratic method yielded disappointing results. However, we believe that this method should be assessed more thoroughly with denser flight paths, and a less variable elevation of the drone above ground, than in the present study. Additionally, field trials should be conducted over open areas such as fields to evaluate the statistical noise, and possible bias, caused by dense canopy. Our results were possibly influenced by the proximity of test transmitters to road edges, so future field trials could be more reliable if transmitters were placed at more varying distances from roads.

Given the aims of this study, it was natural to design the drone search pattern so that it would fly directly over the transmitters. Of course, in real searches for animals wearing transmitters, search patterns would result from a tradeoff between high density, e.g., the mean distance between flight segments, and extent. Maximum detection distance should set a lower limit to search pattern density. We found that signal strength decreased approximately linearly with increasing distance between the source and the drone. Given that we were able to detect few signals at distances further than 500 m, we believe that drones should always fly within 500 m of locations where animals wearing tags are expected, or test transmitters were placed, in the case of calibration studies such as the present one.

Even in cases where drones would be flown in a dense search pattern over large extents, locating tags carried by animals in motion will be more challenging than locating stationary tags. The proximity of an animal body is known to amplify signals [15], but fortunately this effect should not influence location estimation, which is based on *relative* signal strength from one drone location to another. Animals moving fast on the horizontal plane, e.g., birds in flight, would undoubtedly pose the greatest challenge. However, even animals remaining in fixed locations may prove more difficult to locate than fixed tags, if they move (e.g., foraging), because signal strength depends on tag angle relative to the drone, as we found here.

Over the past several years, there has been a dramatic increase in the use of drones to survey and monitor birds by means of optical imaging, including breeding colonies [16–18], wintering and migrating waterbirds [19–21], and individual nest inspections [22–24]. Rapid uptake of these applications has been made possible by the relative maturity, simplicity and accessibility of the requisite technologies, namely sophisticated and user-friendly drone flight control systems combined with compact and very high-resolution digital cameras. However, the applicability of these approaches remains limited to relatively large and/or unconcealed birds, whereas a considerable proportion of species under study or management are small and challenging to locate or directly observe [25]. Thus, it is of continuing interest to develop drone-based solutions to monitor and track birds using alternative sensing methods that do not rely on direct visual observation of subjects, including acoustic sensors [26] and radio telemetry. Such developments would help in addressing more elaborate questions, such as habitat selection. This also extends to non-avian species: Chabot and Bird [6] identified a range of wildlife taxa whose study and monitoring could potentially benefit from drone-based radio-tracking, such as small- to medium-sized mammals including primates, mustelids, rodents and bats, as well as lizards, snakes, land turtles and amphibians.

We believe that the lack of breakthrough in drone-based radio-tracking in the last decade results in part from a focus on more appealing, but more challenging, solutions. One of those is based on an "active" localization system whereby multiple antennas mounted on a drone would enable real-time, onboard triangulation of an animal's location, which in turn automatically adjusts the drone's heading to fly towards the animal and pinpoint its position [6]. Indeed, the complexities involved in bringing this idea to fruition have proven exacting, and the best working prototypes to date still require a human operator at the drone's ground control station to manually determine and transmit flight path adjustments based on real-time feedback from the onboard antenna-triangulation system [7]. In contrast, the approach detailed in our study and explored by few others [27] consists of a "passive" localization system whereby the drone executes a preprogrammed flight path over an area potentially containing one or more radio-tagged animals. The varying signal strength of a given transmitter received by a single antenna on the drone at multiple locations along its flight path is analyzed post hoc to estimate the animal's location. Although not as compelling a solution as active localization, this passive approach can be much more readily implemented using existing technology (i.e., a drone capable of autonomous waypoint navigation and a simple antenna–receiver–logger system) without needing to develop and integrate sophisticated new gadgetry.

Continuing technological and operational advancements of drones are likely to improve their effectiveness for wildlife radio-tracking going forward. Already, "terrain following" capabilities have now been integrated into most popular drone flight control systems, enabling the aircraft to maintain a constant altitude above ground level in areas of varying relief, and consequently better normalization of the strength of radio signals received from the ground. The flight range of drones is also currently limited both by battery capacity and, often more significantly, airspace regulations that predominantly restrict drone operation to within visual line of sight of operators on the ground [28,29]. This restriction tends to be especially crippling in forests, where tall and dense trees, and hilly topography surrounding ground operators cause them to quickly lose sight of the drone as it flies away. It is therefore promising that regulatory agencies in several countries have recently undertaken more serious considerations of allowing drone operations beyond visual line of sight (BVLOS) under certain conditions, and even begun granting BVLOS approvals in limited cases. Regarding flight endurance, fixed-wing drones can typically remain airborne significantly longer (upwards of an hour) than rotary-wing drones (typically <30 min), but the takeoff and landing space requirements of the former tend to prohibit their use in areas such as forests, whereas the latter feature more versatile vertical takeoff and landing (VTOL). However, a growing number of hybrid VTOL fixed-wing drones have recently begun to enter the commercial market.

## 5. Conclusions

Despite the limited number and extent of drone flights in this study, we were able to obtain detection sets with enough detail to provide an operational assessment of transmitter location errors. We conclude that accuracy in this system was insufficient for high-precision purposes such as finding nests. However, in the absence of a dense array of telemetry towers, the use of drone-borne receivers may be a cost-effective way to enhance the quantity and quality of data, relative to deploying personnel in difficult-to-access terrain.

**Author Contributions:** Conceptualization, A.D., J.A.T., Y.A., D.M.B. Methodology, A.D., J.A.T., Y.A., P.P., D.M.B., D.C.; Validation, A.D., J.A.T.; Formal Analysis, A.D.; Resources, A.D., J.A.T., Y.A., D.M.B., P.P.; Data Curation, A.D.; Writing-Original Draft Preparation, A.D.; Writing-Review & Editing, A.D., D.C., J.A.T., Y.A., D.M.B., P.P.; Project Administration, A.D., J.A.T.; Funding Acquisition, A.D., J.A.T., Y.A.

**Funding:** This research was funded by the Natural Sciences and Engineering Council of Canada grant number 170173 to Desrochers and Environment and Climate Change Canada.

**Acknowledgments:** We thank the personnel at the Forêt Montmorency for their significant support.

**Conflicts of Interest:** The authors declare no conflict of interest. The funders had no role in the design of the study; in the collection, analyses, or interpretation of data; in the writing of the manuscript, and in the decision to publish the results.

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
