# Peer review of "Estimating Wildlife Tag Location Errors from a VHF Receiver Mounted on a Drone"

_drones, doi:10.3390/drones2040044_

Round 1

Reviewer 1 Report

This manuscript demonstrates the utility of using drones to detect birds using radio tracking under remote/rugged settings. This approach could greatly reduce the difficulties of current monitoring of elusive animals under these settings. The manuscript has been compiled well and will be of broad interest.

Author Response

Dear reviewer,

Thank you for your time, we have carefully reread the MS to find and correct minor language issues.

Best,

André Desrochers

Reviewer 2 Report

Nice work ! Seems that there is room for development.

Based on your experience, which are the sources of errors?

Are related to the canopy structure (thus you need more powrfull antennas - transmitters)?

Have you run experiments on open fields to compare the results? 

Author Response

Dear reviewer,

Thank you for your time.

We have not flown the drone over open areas, but that surely would be a relevant thing to do. We added the following sentence early in the Discussion: "Additionally, field trials should be conducted over open areas such as fields to evaluate the statistical noise, and possible bias, caused by dense canopy".

We have carefully reread the MS to find and correct minor language issues.

Best,

André Desrochers

Reviewer 3 Report

The study by Desrochers et al. presents data that will be interesting to a wide range of biologists in study systems around the globe. The data are much needed and fill a large gap in our current understanding of the use of drones in ecological research. In addition to providing data related to the accuracy of position estimates made by drone detection of VHF transmitters, the authors also include information on the orientation of transmitters that adds value for practitioners. I believe that this will be a very well received study, although I think there are two main areas where the manuscript can be improved.

The last paragraph of the methods section needs substantially more detail. A clearer description of the ‘maximum strength method’, including specifically naming it this in the methods, is needed. Similarly, the single sentence describing the quadratic method is not enough. I have an intuition as to how the analysis was completed, but I am in no way certain I understand it correctly and definitely could not replicate it based on the current level of detail. As I point out for line 121, further clarification of the method may indicate that a random effect is needed to account for variation in signal strength among individual transmitters.

The discussion needs substantial re-focusing to ensure that adequate context and interpretation of the results is achieved. At present, only the first two sentences of the discussion are an interpretation of the actual results of this study. The authors have missed great opportunities to provide added details about key aspects of this study that would be of more interest to practitioners in this field than a discussion of the limitations of competing technologies and the suitability of fixed-wing (which were not tested here) compared to rotary wing drones. This discussion could be made greatly more relevant by addressing some of the following:

-the quadratic method returned usable results in only 6 out of 35 cases, and furthermore, a seventh case had an error of more than 3,000 m. The authors clearly need to address why the outcomes of this analysis were so poor, and potentially discuss whether they believe another method needs to be developed to improve estimate accuracy (e.g. repeat transects in quick succession). Similarly, the authors could make a recommendation as to whether they thought the maximum strength method or the quadratic method should be the favoured method.

-The strength of the signal declines with distance to transmitter following a consistent relationship. Could the authors include a discussion as to the maximum distance that a transmitter could be detected based on their findings? This would have implications for how wide drone transects may need to be spaced to ensure adequate coverage of a study area.

-For all but four transmitter placements, the drone flew almost directly over the top of the transmitter (and the minimum distance between the drone and transmitter was usually vastly less that the estimated position error). The authors should discuss whether flying almost directly over the transmitter is really feasible in studies of free-roaming passerines, and how this is likely to influence the accuracy estimates if the method was implemented in real life on wild birds.

-The authors could discuss how the habitat (e.g. mature forest, mid-successional, early-successional) influenced accuracy and also some estimate of canopy density could be given to give researchers an indication as to how this study would generalise to their study system.

-The position of transmitters in this study are all very close to road ways. Is this likely to influence results i.e. does the canopy gap created by the road make detection more accurate? An analysis of accuracy as a function of distance from road/canopy gap of the transmitter could be informative.

-The finding that transmitter orientation (up, horizontal, down) is not discussed at all. How does the typical attachment method for passerine birds compare to these orientations and does the finding that a transmitter pointing up results in greater signal strength have implications for optimal harness attachment?

-Accuracy goes down when points beyond the minimum convex hull were excluded. This isthe opposite effect to what was intended. Why might this be? Is it chance alone, or does it suggest that having more points that could potentially be the closest one generates random noise that is not so much of a factor when all of the points are on one side of the true location?

Specific Comments

Title: Why have you decided to limit the context to just birds? Surely these findings are relevant to the study of organisms other than birds.

Introduction

L45 ‘remain predominantly used’ This is not the case in all fields of telemetry. For instance, they are rarely used in seabird research even on small birds due to the difficulties in relocating them over large expanses of water. I suggest changing this to ‘remain the predominantly used tracking technology in studies of small songbirds and many other terrestrial applications.’

Materials and Methods

L83 The transmitters are referred to as ‘dummy’ transmitters when in fact they are functional transmitters. I suggest removing the word dummy. This can be change throughout the manuscript (e.g. figure 3 caption, line 120 etc.)

L109 The reference for this sentence is inappropriate. This is the software that was used to do the analyses, when it should be a citation of a paper using the analysis method referred to in this sentence. Add a suitable reference here and add an additional sentence stating ‘All analyses were conducted using the statistical software R version XXXX [14].’

Results

L112 ‘A total of 652 of those detections were within 350 m of the drone’ This does not really tell us anything because, based on Figure 2, your flight paths were almost entirely within this distance to the transmitter locations. Therefore, this finding isn’t saying anything about the maximum distance that someone may be able to detect a transmitter from, rather it is capped by the maximum distance you flew from your transmitters in this study.

L118 The data in Figure 2 show a spread of points on the x axis up to ~333 m. However, the second sentence of the results clearly indicates that data exist for distances in excess of 350 m (there should be 17 points >350 m). What happened to these data?

L121 If the same five individual transmitters were used in each of the seven flights, these 35 detection sets are not independent datum points. In this case, transmitter ID should be included as a random effect in the analysis to control for inter-transmitter biases (e.g. different signal strength as a result of variations in battery depletion).

L121 Please clarify if all five transmitters were detected on each of the seven flights or if for some flights one or more individual transmitters were not detected at all. This has implications for how much search effort a biologist would need to do to ensure that they were certain their tracked individual was not in that area.

L142 Please state that the dots are the transmitter locations and the other end of the line is the position of the drone (or the other way around if that is the case).

Discussion

L145 As for the title, I do not see why you have restricted the applicability of your results to studies of birds. Findings would be relevant for mammal studies and potentially other taxa too.

Author Response

Dear reviewer, thanks for your in-depth review and very thoughtful comments. We attach a detailed response to your comments, and hope you will find our response satisfactory.

A. Desrochers.

Round 2

Reviewer 3 Report

The changes made by Desrochers et al. to the manuscript 'Estimating bird tag location errors from a VHF receiver mounted on a drone' Have greatly improved the value of this work. The authors have done a fantastic job of incorporating my recommendations from the previous round of review, and I now believe that the research is presented in a balanced way that not only highlights the positives or the study, but also successfully acknowledges the limitations of the study design. Readers will now find the manuscript more useful for generalising to their study system and planning future projects. I now think that this manuscript is suitable for publication.

I have only two minor changes to wording:

L185 insert ‘fly’ or ‘pass’ before 'directly over the transmitter'

L189 Change ‘Given that we were able to detect few signals further from 500 m’ to ‘Given that we were able to detect few signals at distances further than 500 m from test transmitters’

Besides these very minor changes, the manuscript can be accepted in its present form.

Author Response

Dear reviewer, Thanks for the very positive comments, we appreciate a lot. Of course, we will proceed with the edits.

André Desrochers, on behalf of all co-authors.